# Longitudinal Analysis of Plantar Pressures with Wear of a Running Shoe

**DOI:** 10.3390/ijerph17051707

**Published:** 2020-03-05

**Authors:** Elena Escamilla-Martínez, Beatriz Gómez-Martín, Lourdes María Fernández-Seguín, Alfonso Martínez-Nova, Juan Diego Pedrera-Zamorano, Raquel Sánchez-Rodríguez

**Affiliations:** 1University Clinic of Podiatry CPUEX, University of Extremadura, Centro Universitario de Plasencia, Avda, Virgen del Puerto 2, 10600 Plasencia, Spain; escaelen@unex.es (E.E.-M.); bgm@unex.es (B.G.-M.); rsanrod@unex.es (R.S.-R.); 2Nursing Department, University of Extremadura, 10071 Cáceres, Spain; jpedrera@unex.es; 3Physiotherapy Department, University of Sevilla, 41004 Sevilla, Spain; lfdez@us.es

**Keywords:** running shoes, baropodometry, plantar pressures, Biofoot/IBV^®^, running

## Abstract

Running shoes typically have a lifespan of 300–1000 km, and the plantar pressure pattern during running may change as the shoe wears. So, the aim of this study was to determine the variation of plantar pressures with shoe wear, and the runner’s subjective sensation. Maximun Plantar Pressures (MMP) were measured from 33 male recreational runners at three times during a training season (beginning, 350 km, and 700 km) using the Biofoot/IBV^®^ in-shoe system (Biofoot/IBV^®^, Valencia, Spain). All the runners wore the same shoes (New Balance^®^ 738, Boston, MA, USA) during this period, and performed similar training. The zones supporting most pressure at all three study times were the medial (inner) column of the foot and the forefoot. There was a significant increase in pressure on the midfoot over the course of the training season (from 387.8 to 590 kPa, *p* = 0.003). The runners who felt the worst cushioning under the midfoot were those who had the highest peak pressures in that area (*p* = 0.002). The New Balance^®^ 738 running shoe effectively maintains the plantar pressure pattern after 700 km of use under all the zones studied except the midfoot, probably due to material fatigue or deficits of the specific cushioning systems in that area.

## 1. Introduction

Physical exercise, together with other healthy lifestyles, is the main public health strategy aimed at maintaining optimal health and preventing disease. Adequate foot health makes it possible to do any exercise carried out under load. Currently, distance running is a widely extended and increasingly popular physical activity aimed at improving both physical and mental health [1]. There are several factors that influence running, including technique [2,3,4], age [5,6], and the type and inclination of the running surface [7,8,9]. With respect to the foot, the most important factors that can influence distance runners are the running shoes used [9,10,11] and the runner’s foot type [12,13].

The fundamental objectives of running shoes are stability [14] and cushioning [15,16], especially under the heel since this receives the initial impact [17]. At each heel strike, the runner is subjected to ground reaction forces that are approximately 1.5–3 times the body weight [15,18]. These forces cause vibrations or shock waves that are transmitted from the bones of the foot to the rest of the body, and which may be the source of injuries related to the continual practice of running [19]. The shoe can reduce overall impact forces by up to 35% [20,21], but can lose this capacity as the midsole materials wear away [22].

Each particular running shoe model has an approximate average life determined by the use made of it (terrain, speed, running style,…) and by the user’s personal characteristics (weight, height, foot type,…). It would be desirable to quantify the useful life of running shoes in a personalized way so as to estimate when a model has lost its cushioning properties. However, there have been only a few longitudinal baropodometric studies that have examined the variation of plantar pressures with the wear of the shoe. Examples are those of Wegener et al. [21] who compared plantar pressure patterns with different models of training shoes, of Wiegerinck et al. [23] with training versus competition shoes, and of Clinghan et al. [20] who compared different priced shoes. In a study of plantar pressures with various models of running shoe, Dixon [24] evaluated two of the models at the beginning and at 800 km (500 miles) of use. Verdejo and Mills [25] found a 100% increase in peak plantar pressure after 500 km of use. Kong et al. [26] looked at the differences between new and used shoes, noting among other things that the support phase was lengthened with used shoes.

Despite the aforementioned important contributions, and even though the shoe is one of the most important factors influencing the distribution of plantar pressures, to the best of our knowledge, there has been no longitudinal study evaluating the evolution of plantar pressures throughout the useful life (start, mid-life, and end of useful life) of a training shoe during a sports season or cycle. It would also be interesting to inquire into the runners’ perceived comfort during that process. The objective of the present study was therefore to determine the variation of plantar pressures with the wear of running shoes during three different periods, and to examine its relationship with the cushioning that the runners perceive.

## 2. Materials and Methods

### 2.1. Subjects

The sample consisted of 33 male distance runners (mean age 39.3 ± 6.7 years, range 29–56 years; height 1.72 ± 0.08 m; weight 70.7 ± 9.1 kg; and Body Mass Index (BMI) 23.7 ± 2.3 kg/m^2^ and a weekly mileage of 53 ± 12.3 km) who had been distance running as amateurs for at least 5 years. All the subjects were heel strikers, as they themselves declared and it was confirmed visually. All were screened by means of an interview and physical examination for obvious foot or gait abnormalities. The inclusion criteria were: (a) over 25 years of age; (b) running at least 35 km/week; (c) sufficient physical capacity to be able to run at a speed of 12 km/h for one hour. The exclusion criteria were: (a) significant foot or lower-limb abnormalities, history of leg length discrepancy, foot surgery, fractures, or pain in any foot region; (b) any subject who, for some reason, did not run on the indicated firm ground type of running surface, did not meet the established weekly distance in kilometers, or who suffered some injury. Following approval of the research design by the Research Ethics Committee of the University of Extremadura (id: 41/2010), all participants signed their informed consent to participate in the study.

### 2.2. Protocol

All the runners logged daily the kilometers they had run on hard-ground country tracks or asphalt. They were provided with the same model of shoe (New Balance^®^ 738, Boston, MA, USA) to wear for this regular training only. This is a neutral midrange shoe, with a manufacturer’s expected useful life of about 700–800 km. This model was selected as being adequate for the runners, and was available at the time of the study in a full range of sizes.

The participants responded to a questionnaire (dichotomous and Likert items) at the end of the study to score from 1 to 10 the cushioning that they perceived in different zones of the foot, and gave an overall rating to the shoe.

### 2.3. Plantar Pressures Measurement

Pedobarometric measurements were made with the Biofoot/IBV^®^ (Instituto de Biomecánica de Valencia, Valencia, Spain) instrumented insole system (Figure 1). It consists of a pair of flexible insoles with 64 piezoelectric sensors (0.5 mm thickness, 5 mm diameter). Data is sent by digital telemetry from the amplifier to be logged on a computer and then processed by software that shows the plantar pressure (kPa), contact time (s), and cadence (steps/minute) parameters. The digital telemetry system has a range of 200 m with sampling rates between 50 and 250 Hz. The system has been shown to be reliable [27].

Readings were made on three occasions: when the shoe was new (Measurement 1), at 350 ± 10 km (Measurement 2), and at 700 ± 10 km (Measurement 3). The measurements were made on a treadmill running at a speed of 2.77 m/s (which is comfortable for most runners). Data was logged at a rate of 250 Hz, ideal for running measurements [28]. After allowing the subject time to get used to running on the treadmill, measurements were made over 10 s, sufficient to analyze 8 to 12 strides involving both feet. The maximum peak pressure (MPP), expressed in kilopascals (kPa), was determined under the following areas of the foot: rearfoot, midfoot, forefoot, and medial and lateral zones (Figure 2), selected automatically by the software, together with the contact time and the cadence.

### 2.4. Statistical Analysis

A Kolmogorov–Smirnov test showed the data to be normally distributed, allowing parametric statistical tests to be used. Measurements 1, 2, and 3 of the MPP are expressed by their descriptive statistics (mean ± standard deviation). Their differences were studied using Student’s t-test for paired samples and a multivariate analysis of variance (Pillai’s trace). All statistical analyses were performed using the software package SPSS vn 15.0 (SPSS, Chicago, IL, USA). The significance level was taken to be *p* ≤ 0.05.

## 3. Results

### 3.1. Plantar Pressures

In the three measurements (Table 1), the greatest pressures recorded were on the medial (inner) column of the foot (Measurement 3: 1052.7 ± 543.4 kPa) and the forefoot (Measurement 3: 998.5 ± 564.8 kPa). The lowest MPPs were on the rearfoot and midfoot. There was an increasing trend over time in all the MPP measurements and foot zones studied, with a significant increase in the midfoot zone (m1: 387 kPa; m2: 450 kPa; m3: 590 kPa; *p* = 0.027; Table 1). The Pillai’s trace test confirmed the increasing trend of plantar pressures in the midfoot over the three measurements (*p* = 0.008).

### 3.2. Perceived Cushioning

The runners scored the shoes’ cushioning with a mean of 6.2 ± 1.8 points for the heel zone. This was followed by the forefoot cushioning (5.5 ± 1.6), while the lowest score was for the midfoot (5.2 ± 1.4 points). The overall score given to the shoe was 6.1 ± 1.4 points. With respect to the relationship between MPP and the perceived cushioning on the one hand, and the overall score given to the shoe on the other (Table 2), there were negative correlations corresponding to the midfoot area (*r* = −0.513, *p* = 0.002; and *r* = −0.417, *p* = 0.016; respectively).

## 4. Discussion

The purpose of the study was to determine the variation of plantar pressures with shoe wear as well as the runner’s subjective sensation. The findings showed that, during the training season, there was a significant increase in pressure on the midfoot, and the runners with the greatest peak pressures on that area being those who felt the worst cushioning.

Muscle fatigue produced by continual running, together with the normal tendency of the foot to adapt to the ground by pronation [29,30,31] and the transfer of loads to the great toe [12], may be the causes of the greater pressures measured under the medial zone of the foot (Table 1). In addition, runners’ feet are usually more pronate [32], and the fact of using a neutral shoe without pronation control allowed the foot to move freely.

The second greatest MPPs corresponded to the forefoot (Table 1). This may be due to the high load it supports in the propulsion phase of running gait. Once the heel lifts off from the ground, the forefoot bears the entire load of the body, as has been determined by the values measured in other studies [23,33]. Additionally, the rapid transition of support from rearfoot to forefoot during the dynamics of running gives these areas particular biomechanical relevance at the expense of the relevance of the midfoot [21,34].

With respect to shoe wear with use, the second and third sets of measurements show that the cushioning properties of the shoe have gradually been lost between 350 and 700 km. Other studies [21] have found that when a new shoe is first used, there is a 12% to 17% increase in the midfoot loads. One would therefore deduce that this zone of the foot is likely to undergo the greatest variation in the peak pressures it supports.

These results differ in part from those found in the literature. Verdejo and Mills [25] report a 100% increase in peak plantar pressures after 500 km of use. However, their methodological approach differed from ours since they used the same model shoe for only 3 runners who covered 500, 700, and 725 km, respectively, and, indeed, those authors noted that there had yet to be published any longitudinal study of changes in plantar pressures with the progressive use of sports shoes.

With respect to the rearfoot, forefoot, and inner and outer zones, the New Balance^®^ 738 shoe would seem to be effective in essentially maintaining the original maximum pressures up to 700 km of use. An avenue for future research would be to examine the case when longer distances have been run.

The increased plantar pressures in the midfoot may have various causes, of which four would seem to be the most likely. The first could be excessive crushing of the cells of the midsole material. Verdejo and Mills [25] found cells of ethyl vinyl acetate (EVA) foam in running-shoe midsoles to flatten with material fatigue, being damaged and broken at 750 km of use, thus explaining the known reduction in the shoes’ cushioning capacity [35]. In the present case, although the N-ERGY© midsole system of the shoes our subjects used does not consist of EVA foam but of different polymers, it may behave similarly with fatigue of the material and the consequent reduction in absorption of impacts on the midfoot. The second cause may be the absence of specific cushioning systems in this area of the shoe, and indeed, in light of the present results, such systems might be necessary in running shoes. Shoes with specific motion control systems in the midfoot could provide another result, due to increasing ground contact times [16] or to different muscular activation [36,37]. The third could be the subjects’ type of foot since, for instance, excess pronation [33] or pes cavus [38] can influence the pattern of pressures. The fourth cause could be that plantar pressures measured in a training shoe could be similar to those in a racing shoe even though this allows the foot more freedom of movement [39]. One would assume that, with both types of shoe, the materials become less effective with use, resulting in similar increases in plantar pressures. In this regard however, Wiegerinck et al. [23] and Dixon [24] concluded that running with competition footwear involves higher peak plantar pressures than with training footwear.

With respect to published longitudinal studies on the wear of running shoes, changes are observed in the map of plantar pressures after 220 km of use with different models of footwear [40]. At over 800 km (500 miles) of use, there are differences in the pressures according to which system is used to cushion the foot, with the gel model being more effective than EVA foam in the midfoot area [24].

In summary, plantar pressures change according to the model of running shoe used and the wear it has undergone. Since it has been shown that footwear is one of the factors that helps prevent sports injuries [41], it is necessary to respect its average life because if used for too long, there will be an increased risk of sports injuries [42], including stress fractures [39], muscle overload, and ligament or tendon strains.

The results regarding the perceived cushioning of the shoe and its overall score (Table 2) showed that the runners who perceived the poorest cushioning in the midfoot zone were those with the highest MPPs in this area. This highlights the importance of the subjective perception of comfort or discomfort on the part of the runners themselves, and would justify the incorporation of specific cushioning systems allowing greater comfort and security in this part of the shoe, thereby decreasing the risk of injury.

Various works in the scientific literature reflect the importance of comfort in running shoes [43], but only some of them describe methods to reliably and repeatedly evaluate comfort, such as by means of a visual analogue scale [21].

### Limitations of the Study

The present study has some limitations and the results should be interpreted with caution. The main limitation is in the lack of consideration of the participants’ type of foot, with the findings instead being based upon a specific model of shoe.

## 5. Conclusions

The New Balance^®^ 738 running shoe effectively maintains the plantar pressure pattern after 700 km of use in the rearfoot, forefoot, medial, and lateral zones, but fails in the midfoot zone, possibly due to material fatigue or to specific deficits of the cushioning systems in this area. While the present results cannot be extrapolated to all training shoes for distance running, they should serve as orientation and referents. The movement of pronation and the loads generated by the foot in the propulsion phase in running, as measured on a treadmill, could explain why the medial zone and the forefoot support the greatest loads on the foot. The subjective feeling of comfort may serve as a referent in signaling the beginning of incipient wear inside the running shoe.

## Figures and Tables

**Figure 1 ijerph-17-01707-f001:**
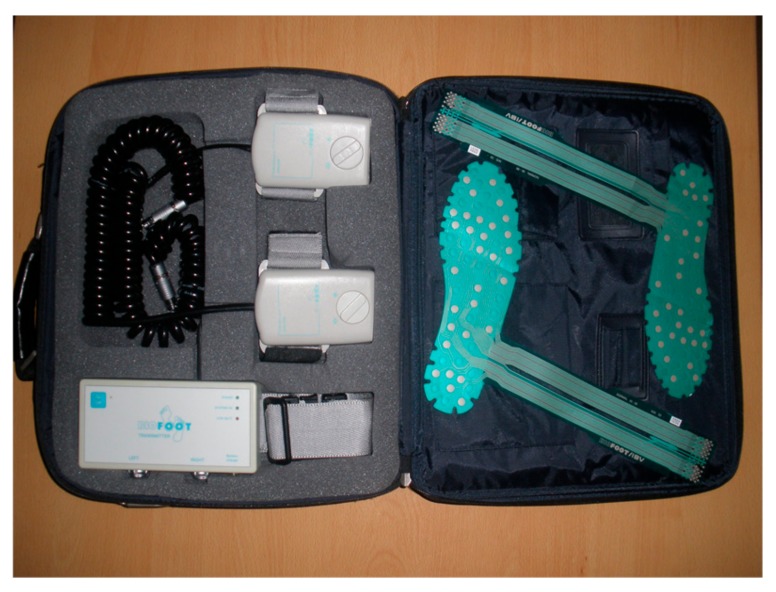
Biofoot in-shoe system.

**Figure 2 ijerph-17-01707-f002:**
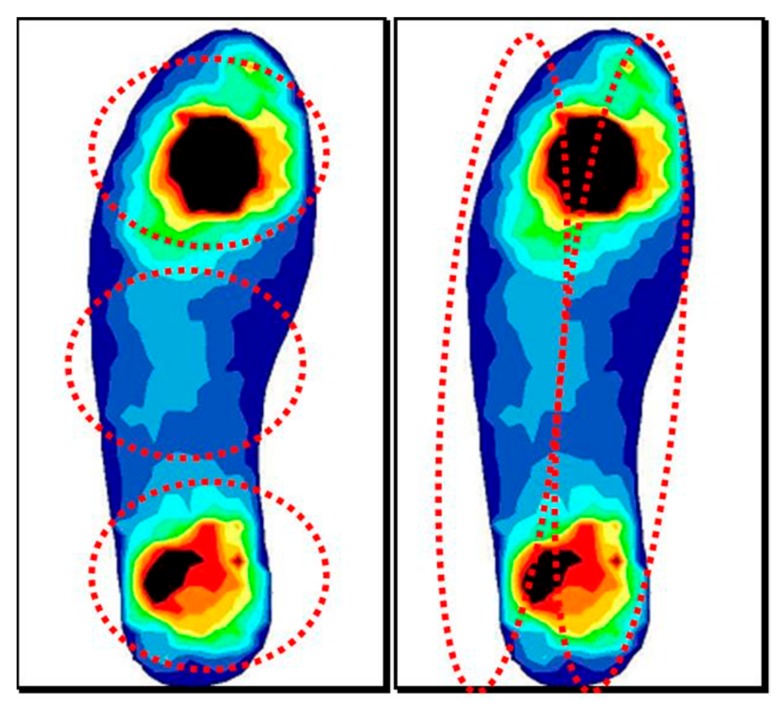
Regions of interest analyzed. Rear, mid, and forefoot. Inner and outer foot.

**Table 1 ijerph-17-01707-t001:** Plantar pressures in the three stages of the study.

Variable	Measurement 1	Measurement 2	Measurement 3	*p*
	Mean ± SD	Mean ± SD	Mean ± SD	
Contact time (s)	0.32 ± 0.08	0.3 ± 0.08	0.33 ± 0.07	0.582
Cadence (steps/min)	161.9 ± 30.8	148.2 ± 30.6	158.3 ± 37.6	0.610
Lateral peak pressure (kPa)	766.1 ± 465.2	787.6 ± 497.8	838.1 ± 508.6	0.501
Medial peak pressure (kPa)	969.2 ± 504.4	1023.8 ± 478.8	1052.7 ± 543.4	0.476
Rearfoot peak pressure (kPa)	639.9 ± 506	767.7 ± 438.4	773.9 ± 656.9	0.238
Midfoot peak pressure (kPa)	387.8 ± 205.3	450.3 ± 297.4	590 ± 457.6	0.027
Forefoot peak pressure (kPa)	884.6 ± 554.7	913.6 ± 548.2	998.5 ± 564.8	0.329

**Table 2 ijerph-17-01707-t002:** Correlations between perceived maximum peak pressure (MPP) and cushioning.

Variables		*r*	*p*
**N = 33**			
MPP rearfoot	Cushioning rearfoot	−0.265	0.136
MPP midfoot	Cushioning midfoot	−0.513	0.002
MPP forefoot	Cushioning forefoot	−0.287	0.106
MPP rearfoot	Overall score	−0.332	0.059
MPP midfoot	Overall score	−0.417	0.016
MPP forefoot	Overall score	−0.301	0.083
MPP maximum peak pressure; Pearson correlation.			

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
