# Peer review of "Longitudinal Analysis of Plantar Pressures with Wear of a Running Shoe"

_ijerph, 2020, doi:10.3390/ijerph17051707_

Round 1

Reviewer 1 Report

This manuscript entitled “longitudinal analysis of plantar pressures with wear of a running shoe” intended to analyze the difference of plantar loadings during the three different training periods. But there are still some problems, for instance, the method used in this experiment and the English writing that can’t up this review to a publishing level. Some suggestions are listed in the specific comments below.

Specific comments:

  1. In the abstract section, for the keyword ‘sport shoe’, change it to ‘running shoes’.
  2. In the introduction section, line 37, is it the GRF 2-5 times of the body weight?

The reference from Daniel E. Lieberman et al (2010): ‘rearfoot runners must repeatedly cope with the impact transient of the vertical ground reaction force, an abrupt collision force of approximately 1.5-3 times body weight.’

  1. Line 42, change ‘a certain approximate average life’ to ‘an approximate average life’.
  2. Line 45-46, In this sense, various pedobarometric studies on sports shoes can be found in the literature [18,19,21]. Same as line 50-51.
  3. Line 53-54, for running shoes, is it a real 100% in peak pressure after the use of 500km.
  4. Line 57, ‘to the best of our knowledge/ to the best of the authors’ knowledge’ rather than ‘to the best of the author’s knowledge’.
  5. Line 56-64, combining these two paragraphs into one paragraph. Line 63, with the wear of running shoes during three different periods.
  6. In the methods section, change ‘for at least the past 5 years’ to ‘at least 5 years’. Line 69, remove ‘when running’.
  7. Line 82-84, Giving more details for this questionnaire.
  8. Line 86, showing the information for this apparatus.
  9. In the discussion section, please discuss the main purpose, method and findings in the first paragraph.
  10. Line 170, change ‘in sum therefore’ to ‘in summary’.

Author Response

Thank you for your kind and very helpful comments. I have looked through the manuscript following your recommendations and suggestions. I will show to you the changes that we have made due to your review. The changes in the text made due to your revision are underlined in yellow

Reviwer 1

This manuscript entitled “longitudinal analysis of plantar pressures with wear of a running shoe” intended to analyze the difference of plantar loadings during the three different training periods. But there are still some problems, for instance, the method used in this experiment and the English writing that can’t up this review to a publishing level. Some suggestions are listed in the specific comments below.

Response: The paper has been edited by an English native translator.

Specific comments:

  1. In the abstract section, for the keyword ‘sport shoe’, change it to ‘running shoes’.

Response: This change has been made in the text

  1. In the introduction section, line 37, is it the GRF 2-5 times of the body weight? The reference from Daniel E. Lieberman et al (2010): ‘rearfoot runners must repeatedly cope with the impact transient of the vertical ground reaction force, an abrupt collision force of approximately 1.5-3 times body weight.’

Response: This change has been made in the text

  1. Line 42, change ‘a certain approximate average life’ to ‘an approximate average life’.

Response: This change has been made in the text

  1. Line 45-46, In this sense, various pedobarometric studies on sports shoes can be found in the literature [18,19,21]. Same as line 50-51.

Response: This change has been made in the text, two lines was removed

  1. Line 53-54, for running shoes, is it a real 100% in peak pressure after the use of 500km.

Response: This change has been made in the text

  1. Line 57, ‘to the best of our knowledge/ to the best of the authors’ knowledge’ rather than ‘to the best of the author’s knowledge’.

Response: This change has been made in the text

  1. Line 56-64, combining these two paragraphs into one paragraph. Line 63, with the wear of running shoes during three different periods.

Response: This change has been made in the text

  1. In the methods section, change ‘for at least the past 5 years’ to ‘at least 5 years’. Line 69, remove ‘when running’.

Response: This change has been made in the text

  1. Line 82-84, Giving more details for this questionnaire.

Response: This change has been made in the text. The type of the questions has been added

  1. Line 86, showing the information for this apparatus.

Response: This info has been added in the text, and also a reference of their reliability

  1. In the discussion section, please discuss the main purpose, method and findings in the first paragraph.

Response: This change has been made in the text

  1. Line 170, change ‘in sum therefore’ to ‘in summary’.

Response: This change has been made in the text

Reviewer 2 Report

Dear authors.

I think that this is a curious study. I have some minor comments for authors:

Line 70: Inclusion criteria: over 25 years of age. Why was over 25 years and not 18 years?

Line 82: The participants responded to a questionnaire at the end of the study to score from 1 to 10 the cushioning that they perceived in different zones of the foot. Is the questionnaire validated? I think that is necessary more information about this questionnaire.

Line 116. I think that is a misprint. Percieved cushioning: Perceived

Line 128. Authors say : runners' feet are usually more pronate [29]. Have you study the foot type of our study population and the possible influence in the plantar pressures? As author have comented in line 158: excess pronation or pes cavus can influence the pattern of pressures.

Finally I am struck by this study has not limitations. Authors should thinking about this?

Author Response

Thank you for your kind and very helpful comments. I have looked through the manuscript following your recommendations and suggestions. I will show to you the changes that we have made due to your review. The changes in the text made due to your revision are underlined in green.

Reviewer 2

I think that this is a curious study. I have some minor comments for authors:

Line 70: Inclusion criteria: over 25 years of age. Why was over 25 years and not 18 years?

Response: Because athletes had to be an experience of 5 years running being adults. We wanted experienced runners, adapted largely to the sport. If they started to running in 18-20 years, 5 years later the age will be around 25.

Line 82: The participants responded to a questionnaire at the end of the study to score from 1 to 10 the cushioning that they perceived in different zones of the foot. Is the questionnaire validated? I think that is necessary more information about this questionnaire.

Response: The questionnaire is not validated (since to our knowledge is not a validated one for cushioning in running shoes). A new info about the type of questions has been added to the text.

Line 116. I think that is a misprint. Percieved cushioning: Perceived

Response: This change has been made in the text

Line 128. Authors say : runners' feet are usually more pronate [29]. Have you study the foot type of our study population and the possible influence in the plantar pressures? As author have comented in line 158: excess pronation or pes cavus can influence the pattern of pressures.

Response: This issue has been added in the limitations of the study section, because the type of the foot was not assessed.  

Finally I am struck by this study has not limitations. Authors should thinking about this?

Response: This section has been added in the text

Reviewer 3 Report

General comments

Although the shoe is one of the most important factors influencing the distribution of plantar pressures, little study has been conducted to evaluate the evolution of plantar pressures throughout the useful life of a training shoe. The purpose of the present study was to determine the variation of plantar pressures with the wear of the shoe and the runner's subjective sensation. Although this study is interesting, some aspects must be improved, particularly in the methods and results sections.

Specific comments

Line 67: Lack of subjects’ information, e.g., height, weight, weekly mileage, etc.

Line 68: How did you determine the footstrike pattern?

Line 80: the rationale for choosing this shoe (New Balance® 738)

Line 86: need more details regarding the Biofoot/IBV® instrumented insole system, e.g., manufacturer, country, sensor type, the number of sensors, range, etc.

Line 90: 250 mHz? or 250 Hz?

Line 93: how did you determine the areas of the foot (rearfoot, midfoot, forefoot, and medial and lateral)?

Table 1: Lack of units; indicate the significant differences between which two measurements.

Lines 112-113: please also indicate the significant differences between which two measurements.

Line 184: Lack of limitation part of the study.

Besides, there are grammatical errors and a thorough language review is necessary. Furthermore, the following references seem to be important (one in plantar pressure, the other in running technique), and are suggested to add in the revised manuscript:

- Fu et al. (2015). Surface effects on in-shoe plantar pressure and tibial impact during running, Journal of Sport and Health Science, 4(4): 384-390.

- Huang et al. (2013). Segment-interaction and its relevance to the control of movement during spring. Journal of Biomechanics, 46(12): 2018-2023.

Author Response

Thank you for your kind and very helpful comments. I have looked through the manuscript following your recommendations and suggestions. I will show to you the changes that we have made due to your review. The changes in the text made due to your revision are underlined in blue.

Reviewer 3

Although the shoe is one of the most important factors influencing the distribution of plantar pressures, little study has been conducted to evaluate the evolution of plantar pressures throughout the useful life of a training shoe. The purpose of the present study was to determine the variation of plantar pressures with the wear of the shoe and the runner's subjective sensation. Although this study is interesting, some aspects must be improved, particularly in the methods and results sections.

Specific comments

Line 67: Lack of subjects’ information, e.g., height, weight, weekly mileage, etc.

Response: This info has been added in the text

Line 68: How did you determine the footstrike pattern?

Response: It was do it visually and reported by the athletes. This info has been added in the text.

Line 80: the rationale for choosing this shoe (New Balance® 738)

Response. This running shoe was provided by the manufacturer, as a middle range running shoe. This model was selected as being adequate for the runners of the sample and it was disposable in the moment of the study in all range of sizes. Info added to the text.

Line 86: need more details regarding the Biofoot/IBV® instrumented insole system, e.g., manufacturer, country, sensor type, the number of sensors, range, etc.

Response: This info has been added in the text, and also a reference of it reliability.

Line 90: 250 mHz? or 250 Hz?

Response: 250 Hz, This info has been corrected in the text

Line 93: how did you determine the areas of the foot (rearfoot, midfoot, forefoot, and medial and lateral)?

Response: The areas was determined by the software of the system. This info has been corrected in the text

Table 1: Lack of units; indicate the significant differences between which two measurements.

Response: This info has been added in the text

Lines 112-113: please also indicate the significant differences between which two measurements.

Response: This info has been added in the text

Line 184: Lack of limitation part of the study.

Response: This section has been added in the text

Besides, there are grammatical errors and a thorough language review is necessary.

Response: An English native translator revised and corrected the text to accurate the language.

Furthermore, the following references seem to be important (one in plantar pressure, the other in running technique), and are suggested to add in the revised manuscript:

Fu W, Fang Y, Ming D, Liu S, Wang L, Ren S, Liu Y. Surface effects on in-shoe plantar pressure and tibial impact during running. J Sport Health Sci 2015;4(4): 384-390.

Huang L, Liu Y, Wei S, Li L, Fu W, Sun Y, Feng Y. Segment-interaction and its relevance to the control of movement during spring. J Biomech 2013;46(12):2018-2023.

Response: This references has been added to the text

Round 2

Reviewer 1 Report

Authors have made a good revision, it is suitable to be published now. 

Author Response

Reviewer comments.

Authors have made a good revision, it is suitable to be published now. 

Response: Thanks a lot for your comments

Kind regards

Reviewer 3 Report

Well done. Only one question: regarding "Line 68: How did you determine the footstrike pattern?" the authors responded, "...This info has been added in the text.". However, I did not find it. 

Author Response

Reviewer comments.

Well done.

Response: Thanks a lot for your comments

Reviewer comments.

Only one question: regarding "Line 68: How did you determine the footstrike pattern?" the authors responded, "...This info has been added in the text.". However, I did not find it. 

This info was added in the line 71 (underlined actually in red): ´as they themselves declared and was confirmed visually´

Kind regards
